# Shielding Effectiveness of Unmanned Aerial Vehicle Electronics with Graphene-Based Absorber

Roman Kubacki * , Rafał Przesmycki  and Dariusz Laskowski

Faculty of Electronics, Military University of Technology, 00-908 Warsaw, Poland;
rafal.przesmycki@wat.edu.pl (R.P.); dariusz.laskowski@wat.edu.pl (D.L.)
* Correspondence: roman.kubacki@wat.edu.pl

**Abstract:** Within this study, we explored the augmented security measures for the electronics of unmanned aerial vehicles (UAVs) within an RF environment. UAVs are commonly utilised across various sectors, and their use as auxiliary platforms for cellular networks, as parallel networks working in tandem with ground-based base stations, holds considerable promise. In this context, ensuring the uninterrupted operation of UAVs is a paramount objective. However, the considerable external electromagnetic interference emitted by existing base stations may jeopardise the functionality of UAV electronics. This could potentially lead to an unintended flight path and a sudden cessation of communication with the operator. To mitigate the detrimental impact of the RF field, we advocate covering the UAV casing with reduced graphene oxide (RGO). The efficacy of RGO's shielding effectiveness (SE) was investigated over a frequency spectrum from 100 MHz to 10 GHz. Our scrutiny of this property was centred around the measurement of scattering matrix coefficients of the unadulterated material—without additives of any kind. Our findings show that this material is a favourable candidate for UAV absorbers due to its low reflection coefficient coupled with its high absorption capacity. The studied absorber ensures an SE value of 25 dB and 30 dB for a 3 mm layer at frequencies of 3.6 GHz (pertaining to the 5G system) and 5.8 GHz (pertaining to LTE), respectively.

**Keywords:** unmanned aerial vehicle (UAV); microwave absorber; reduced graphene oxide

## 1. Introduction

An unmanned aerial vehicle (UAV), commonly referred to as a drone, is an aviation apparatus that does not necessitate an on-board crew for its flight operations. Drones can be remotely piloted or equipped with autonomous flight capabilities. The comprehensive system comprises the UAV with its mission-specific apparatus on board, a ground control station manned by an operator, a communications system bridging the control station with the flight apparatus, and ancillary equipment. The demand for drones is substantial, and their prevalence is expected to increase in the coming years. Drones are increasingly used in various domains such as transport, observation, delivery of health-related items, border control, emergency response teams (fire brigades, police), forest monitoring, farming, geodetic surveys, aerial photography, and even for entertainment purposes. As we look ahead, these unmanned vehicles are set to become central in enhancing wireless communication, serving as support systems or acting in tandem with present-day networks like LTE, 5G, and the forthcoming 6G technologies. Such combinations could notably improve communications in IoT or other forward-looking platforms aimed at intelligent city systems, such as smart streets. Within this framework, a drone can act either as a standalone entity or as a node of a more extensive grid. This would mean that communication pathways would involve both "air-to-ground" and "drone-to-drone" interfaces.

A conventional drone is constructed with a non-metallic frame, which forms the backbone of the entire apparatus. The propellers are affixed to rotors, which extend from the engines which, in turn, are powered by batteries equipped with voltage regulators. A

fundamental component of every drone is the IMU (Internal Measurement Unit) module, which incorporates such components as an accelerometer, a gyroscope, and a barometer. More advanced models are additionally furnished with a compass and a GPS module.

The electronics within UAVs, necessary for executing flight operations and facilitating communication with operators or other drones, are sensitive to electromagnetic interferences. Hence, it is imperative that those electronics are shielded from detrimental electromagnetic (EM) radiation. While battery longevity and short flight times continue to present challenges for drones used as communication networks, this limitation may be mitigated by various operational solutions. In contrast, the vulnerability of drone electronics in heightened electromagnetic fields remains a pressing concern. Fields exceeding the EMC immunity threshold could wreak havoc on drone electronics, potentially leading to erratic flight behaviour or uncontrolled descent. Consequently, elevated EM field strengths could result in incidents stemming from sudden communication disruptions with operators or other drones.

Electromagnetic compatibility (EMC) standards dictate that electronic devices must be resistant to radiation [1]. Specific EMC immunity thresholds, as defined in reference [1], are delineated for distinct device categories. Typical commercial electronic devices, classified under Class 2, possess an EMC immunity threshold of 3 V/m, and drones can be categorised within this bracket. However, the established level of electromagnetic immunity is substantially inferior compared to permissible electromagnetic contamination. Global and European standards suggest that EM fields emitted by mobile base stations can reach levels up to 61 V/m [2–4], a threshold adopted by the majority of European nations. This value is far higher than the mandatory EMC protections, hence manufacturers cannot vouch for their devices' efficacy under such intense radiation.

A straightforward countermeasure against such radiation is to envelop the drone's exterior with a microwave absorber. This solution is especially important for UAVs, given that their electromagnetically permeable shells do little to attenuate incoming RF radiation. Ideal absorbers should exhibit high shielding effectiveness (SE) across a broad frequency spectrum, while also being lightweight (to avoid burdening the UAV) and weather-resistant. Presently, two types of materials—namely ferrites and carbon derivatives—meet these microwave shielding prerequisites. Unfortunately, ferrite absorbers, with an approximate density of 5 g/cm$^3$, are too heavy for practical use drones. The new solutions in shielding technology revolve around metamaterials and metasurfaces [5,6]. Carbon-based absorbers, such as graphite [7–9], with their elevated conductivity, appear promising, but commonly available graphite absorbers must be of considerable volume to be effective, and are inherently fragile. However, graphene-based absorbers, especially those made with magnetically inclined atoms (akin to ferrites) coated with graphene [10–12], possess commendable mechanical and electrical properties. Despite these advantages, the relatively heavy weight of ferrite components that comprise these materials render them unsuitable for smaller UAVs.

This study introduces reduced graphene oxide (RGO) as a prospective microwave absorber. We have chosen to harness the inherent properties of unadulterated RGO, which is one of the most lightweight graphene derivatives. Another compelling attribute of RGO is its relatively low production cost. Our exploration underscores RGO's electrical attributes, deeming it an effective absorber that is well-suited for UAVs. We have gauged the intrinsic properties of RGO, specifically its permittivity and permeability, and subsequently analysed the shielding properties contingent on frequency. Our assessment of RGO's shielding effectiveness in free-space conditions, which presumes drone housing to be transparent, provides insights into both its reflectivity and absorption characteristics.

## 2. Absorber Characteristics—Reduced Graphene Oxide

Graphene is one of several allotropes of carbon, along with other forms such as graphite, fullerenes, and nanotubes. Graphene comprises a single layer of carbon atoms arranged in hexagonal rings [13]. A prevalent variant of this material is graphene oxide, which is characterised by the addition of oxygen groups to the carbon layers. However,

this modification diminishes its conductivity, rendering it less effective in absorbing EM energy. To enhance conductivity, these oxygen groups can be eliminated, producing what is known as reduced graphene oxide (RGO), which boasts conductivity levels comparable to pure graphene [9].

To ascertain the material's constitutive properties, it is imperative to measure the complex values of both permittivity and permeability. Given that RGO's final form is a powder, preparing samples for measurements is challenging and rather cumbersome. As a workaround, the extant literature suggests amalgamating RGO powder with other materials, such as wax, resin, or styrofoam [13–16]. Naturally, the electromagnetic attributes of these composites are profoundly influenced by the relative concentrations of the incorporated fillers. In this investigation, RGO was assessed in its unadulterated form, devoid of additives. A solid sample state was achieved by compressing the powder within a bespoke housing using a 2 kN force. The resultant density of the compressed sample was approximately 1.55 g/cm$^3$, rendering it considerably less dense compared to ferrites.

## 3. Coaxial Line Method of Permittivity and Permeability Measurement

The electromagnetic properties of material can be described in terms of relative complex permittivity and permeability:

$$\varepsilon = \varepsilon' - j\,\varepsilon'' \tag{1}$$

$$\mu = \mu' - j\,\mu'' \tag{2}$$

where $\varepsilon'$, $\mu'$ are the electric and magnetic constants and $\varepsilon''$, $\mu''$ are the electric and magnetic loss factors.

The permittivity and permeability measurements of solid materials are commonly performed using coaxial fixtures. At microwaves, the coaxial line technique is especially recommended for broadband frequency measurements. The method of measuring $\varepsilon$ and $\mu$ based on determining the scattering parameters ($S_{ik}$) of the measured sample is the most popular. In this case, the measured material, having toroidal form, completely fills the cross section of the coaxial line. Such configuration guarantees that the only TEM mode propagates in the line.

Measurements of complex relative permittivity and permeability were carried out using a vector network analyzer. This system consisted of a 7 mm coaxial line equipped with measurement cables and LPC7 connectors. The center conductor of the coaxial line is 3.04 mm in diameter to receive the 50 Ω characteristic impedance of the holder. At first, the empty coaxial line was employed to calibrate the system. After calibration, the measurements were updated by the calibration coefficients. Data on the calibration process and measurement of the sample were acquired using the MultiCal program [17,18]. The system measures the magnitudes and phases of S-parameters of a sample. Taking into ac-count that the measured material is homogeneous, only the $S_{11}$ and $S_{21}$ is enough to characterise the electric and magnetic properties. These parameters have complex values depending on the frequency and thickness of the sample and can be determined as follows [19,20]:

$$S_{11} = \rho\frac{1 - T^2}{1 - \rho^2 T^2} \tag{3}$$

$$S_{21} = \frac{\left(1 - \rho^2\right)T}{1 - \rho^2 T^2} \tag{4}$$

where:

$T = e^{-\gamma\,d}$—transmission coefficient.
$\rho = \frac{\sqrt{\mu_c} - \sqrt{\varepsilon_c}}{\sqrt{\mu_c} + \sqrt{\varepsilon_c}}$—reflection coefficient at the boundary "air–material".
$d$—thickness of the measured sample.

The values of unknown complexes $\varepsilon$ and $\mu$ can be obtained solving the set of Equations (3) and (4). It is possible to calculate the permittivity and permeability using the well-known method proposed by Nicolson, Ross, and Weir (NRW method) [19,20]. In this method, values of complex permittivity and permeability can be determined according to the following formulas:

$$\mu = -j\frac{1+\rho}{1-\rho}\frac{\lambda}{2\pi\,d}\ln\left(\frac{1}{T}\right)$$

(5)

$$\varepsilon = -\frac{1}{\mu}\left[\frac{\lambda}{2\pi\,d}\ln\left(\frac{1}{T}\right)\right]^2$$

(6)

where parameters $T$ and $\rho$ can be also determined using $S_{11}$ and $S_{21}$:

$$T = \frac{S_{11} + S_{21} - \rho}{1 - \rho\,(S_{11} + S_{21})}$$

(7)

$$\rho = X \pm \sqrt{X^2 - 1}$$

(8)

$$X = \frac{1 - V1\,V2}{V1 - V2}$$

(9)

$$V1 = S_{21} + S_{11} \quad V2 = S_{21} - S_{11}$$

(10)

The above relationships allow the determination of the complex values of permittivity and permeability of the RGO samples. The measurements were carried out in the frequency range from 100 MHz to 10 GHz. The obtained values of relative complex permittivity ($\varepsilon'$, $\varepsilon''$) and permeability ($\mu'$, $\mu''$) are presented in Figure 1.

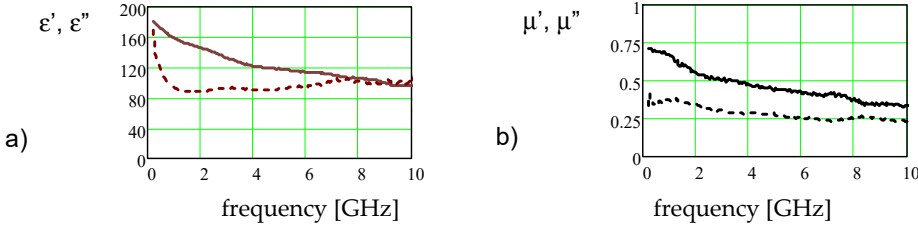

**Figure 1.** (**a**) Permittivity (solid—real part, dotted—imaginary part), (**b**) permeability (solid—real part, dotted—imaginary part).

The complex values of permittivity and permeability of solid state RGO are presented in microwave frequency range. This frequency range includes frequencies emitted by GSM, UMTS, and LTE as well as the 5G mobile system. The electric constant of RGO has a value above 100. The imaginary part of permittivity, being responsible for material loss, has a high value and is quite stable in frequencies above 1 GHz. On the other hand, the magnetic constant assumes values lower than 1, showing the diamagnetic nature of RGO. This phenomenon causes the magnetic field strength to be lower inside the material due to opposite magnetisation. Despite this low value, it is necessary to take into account the magnetic permeability when analysing the absorbed electromagnetic energy in the material. Some-times the permeability of carbon-based materials is neglected, but such simplification leads to incorrect final estimations. In fact, this diamagnetic phenomenon increases the absorption property of RGO, compared to the simplified approach when permeability is taken as $\mu = 1$.

## 4. Shielding Effectiveness

The easiest way to protect drone electronics against harmful radiation is to cover the surface with a broadband absorber. The parameter describing the electromagnetic

protection ability is shielding effectiveness. According to the definition, the shielding effectiveness can be defined as the logarithmic ratio of the magnitude of the incident electric field to the magnitude of the transmitted electric field and can be expressed as follows [21–23]:

$$SE \text{ (dB)} = 20log\left(\frac{E_i}{E_t}\right) \tag{11}$$

where:

$E_i$—incident electric field strength,

$E_t$—electric field strength being transmitted through the material.

There are several factors that determine the effectiveness of shielding, like the following:

- Frequency of the incident EM field;
- Electromagnetic parameters (complex permittivity and permeability);
- Absorber thickness.

With the measured values of permittivity and permeability, it is possible to calculate the shielding effectiveness based on the scattering parameters of the material slab.

$$SE = -10log\left(|S_{21}|^2\right) \tag{12}$$

where $S_{21}$ is the transmitting coefficient of the slab of material with thickness $d$ taken for consideration.

Relationship (12) is a well-known four-terminal reciprocal network equation. This relationship can be also expressed as follows:

$$SE = -10log\left(\frac{|S_{21}|^2}{1 - |S_{11}|^2}\right) - 10log\left(1 - |S_{11}|^2\right) \tag{13}$$

The above modified equation allows the determination of the part of electromagnetic energy being reflected and part of the absorbed energy. The reflection coefficient *LR* and absorption coefficient *LA* are expressed with the following relationships:

$$LA = -10log\left(\frac{|S_{21}|^2}{1 - |S_{11}|^2}\right) \tag{14}$$

$$LR = -10log\left(1 - |S_{11}|^2\right) \tag{15}$$

The total shielding effectiveness is a sum of these components:

$$SE = LR + LA \tag{16}$$

Relationship (16) expresses the electromagnetic losses of all rays propagating inside the material. It also determines the external and internal reflections caused by the absorber. The physical interpretation can be schematically illustrated by analysing the incident ray and the infinite rays propagating inside the material, as illustrated in Figure 2.

Figure 2 shows the EM rays propagating inside the slab of absorber in the free-space condition. Coefficient $S_{21}$ is proportional to the energy transmitted through the material, thus illustrating the shielding properties of the absorber. The EM protection by such material is realised by reflecting the rays and absorbing the energy of all the rays propagating through the absorber. In many applications, the important task is to minimise the reflected part of the incident radiation so as to minimise the *LR*. Such a requirement is especially important for medical apparatuses and other devices when reflected radiation could influence other nearby apparatuses.

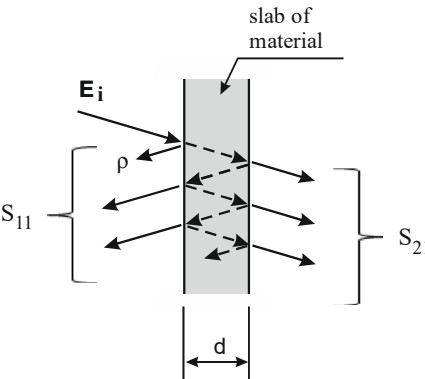

**Figure 2.** Multiple EM ray reflections inside the material.

When the permittivity and permeability are known, it is possible to calculate the coefficients of $S_{11}$ and $S_{21}$ of the slab material with any thickness using Relationships (3) and (4) then, using (14)–(16), the *LR*, *LA* and *SE* can be calculated. The values of these parameters for the two thicknesses, $d$ = 3 mm and 5 mm, are presented in Figure 3.

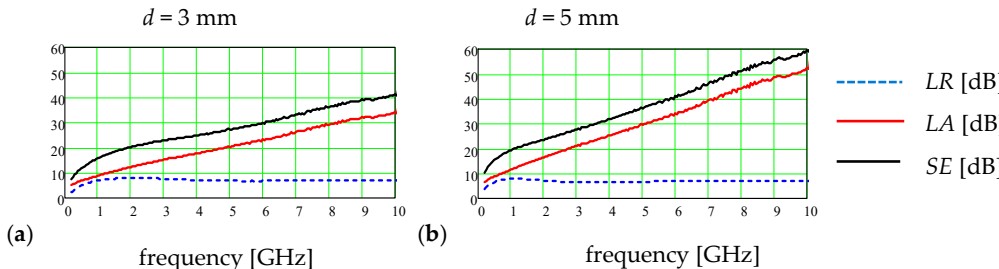

**Figure 3.** Values of of *LR*, *LA*, and *SE* of 3 mm and 5 mm slabs of RGO.

Figure 3 shows the reflection coefficient *LR*, absorption coefficient *LA*, and shielding effectiveness *SE* as a function of frequency, in the range from 100 MHz to 10 GHz. The shielding efficiency was analysed for two example thicknesses of slabs, i.e., 3 mm and 5 mm. Slabs were analyzed in a free-space condition, with the TEM mode incidents perpendicular to the slabs. In a low frequency range, all shielding coefficients have low values and this is a typical property of carbon-based materials. The reflecting coefficient is almost stable as a function of frequency with a value near 7 dB. In fact, the value of *LR* is mainly determined by the reflection coefficient at the boundary "air–material" ($\rho$) because the rays inside the absorber are strongly attenuated so, in this case, the *LR* does not depend on the thickness of slab. In addition, the values of *LR* are significantly lower compared to the absorption coefficient. Such a low value of *LR* guarantees the low reflection property of this material and, from this point of view, the device covered with such an absorber will not disturb the work of other nearby apparatuses. The important part of the shielding efficiency of this material is energy absorption. The function of the absorption coefficient lineally rises with frequency. The components of shielding effectiveness, for a slab of 5 mm, are as follows:

-    At 3.6 GHz (system 5G): *LR* = 6.5 dB, *LA* = 23.5 dB and *SE* = 30 dB.
-    At 5.8 GHz (system LTE): *LR* = 6.8 dB, *LA* = 32.7 dB and *SE* = 39.5 dB.

The UAV electronics covered with an RGO absorber can be effectively protected against harmful radiation. In Table 1, the electric field strength reduced by the absorber layers is presented. In this table, the following parameters were identified. $E_{EMC}$—maximal level of electric field strength, according to EMC, that guarantees that the electronic device works correctly. $E_{in}$—permissible value of electric field strength incidents to a drone. The value of 61 V/m has been considered as a possible RF environment. $E_{SE}$—value of $E_{in}$ reduced by the absorber.

**Table 1.** Values of electric field strength reduced by absorber.

| Frequency [GHz] | $E_{EMC}$ [V/m] | $E_{in}$ [V/m] | SE [dB] | $E_{SE}$ [V/m] |
|---|---|---|---|---|
| | 3 mm thickness | | | |
| 3.6 | 3 | 61 | 25 | 3.4 |
| 5.8 | 3 | 61 | 29 | 0.6 |
| | 5 mm thickness | | | |
| 3.6 | 3 | 61 | 30 | 1.9 |
| 5.8 | 3 | 61 | 39.5 | 0.6 |

With the use of RGO as an absorber, the level of the RF environment can be effectively reduced to the safe level. The safe level, at frequencies emitted by mobile base stations, can be obtained using an even thin layer of 3 mm. In this case, the level of radiation penetrating through the housing of the device is 3.4 V/m at 3.6 GHz and 0.6 V/m at 5.8 GHz. With a thicker layer, it can be much better.

A comparison of the shielding effectiveness of composite material layers with previous studies is shown in Table 2.

**Table 2.** Comparison of shielding effectiveness of material layers.

| Citation | Material | Filler/Structure Modification | Layer Thickness [mm] | Frequency [GHz] | Shielding Effectiveness [dB] |
|---|---|---|---|---|---|
| [24] | Epoxy composite | Carbon nanotube (CNT) | 0.15 | 0.15–1 | 20 |
| [25] | Polymer | Conducting cylindrical inclusions | 10 | 0.1–10 | 10–50 |
| [26] | Woven Fabrics | Au on a polyamide | not specified | 0.03–1.5 | 25–50 |
| [27] | Nylon (PA6) | Silver nanoparticles | membrane | 0.15–3 | 1–2 |
| [28] | Ferrite composition | Co-doped barium hexaferrite | 3 | 2–18 | 35 |
| [29] | Polyaniline | $BaFe_{12}O_{19}$ + carbon nanotube | 4.5 | 8–12 | 37 |
| [30] | Polyvinylidene fluoride | Carbon nanotube | 4 | 8–12.5 | 40 |
| This work | RGO | Pure | 3 | 0.1–10 | 8–40 |
| This work | RGO | Pure | 5 | 0.1–10 | 10–59 |

## 5. Conclusions

The principal objective of this study was to propose an effective microwave absorber, which shields UAV electronics from a harmful RF environment. UAVs have the potential to serve as transportation platforms, enabling the deployment of both mobile–aerial and stationary–aerial cellular networks. Nonetheless, to satisfy the EMC requirements, the incident EM radiation for commercially available devices should not exceed 3 V/m, even though the allowable RF surroundings can reach values up to 61 V/m. The absorber crafted from reduced graphene oxide emerges as a compelling candidate, considering both its electrical and mechanical attributes (notably its lightweight nature and resilience against weather conditions). The investigation of this material was conducted in a frequency range from 100 MHz to 10 GHz in a coaxial line, which ensured the incidence of the TEM field to a slab of material. The shielding effectiveness was subsequently computed for the selected material thicknesses of 3 mm and 5 mm under free-space conditions, with their utility for UAVs being a primary focus. Typically, the material constituting drone housing boasts low permittivity without significant electrical loss, rendering it transparent to the incident EM

field. The assessment of RGO revealed commendable absorption capabilities, exhibiting a minimal reflection coefficient and a satisfactory absorption coefficient in the total shielding effectiveness. A 3 mm layer is sufficiently capable of reducing the maximum permissible RF environment to a level that is deemed safe for drone electronics. Hence, the proposed absorber can reliably safeguard UAV electronics, mitigating the risks of system failures or damage within RF environments.

For the laboratory measurements of permittivity and permeability, pure RGO samples were fashioned into a toroidal shape, which was achieved by compressing the powder within a specialised housing. Presently, efforts are underway to design configurations that facilitate the easy affixation of such absorbers to UAVs.

**Author Contributions:** Conceptualisation, R.K., R.P. and D.L.; methodology, R.K. and D.L.; software, R.P.; validation, D.L.; formal analysis, R.K. and R.P.; writing—original draft preparation, R.K.; writing—review and editing, R.P. and D.L. All authors have read and agreed to the published version of the manuscript.

**Funding:** This work was financed by Military University of Technology under research project UGB/859/2023 on "Estimation of electromagnetic compatibility between DVB-T2 and 4G/5G in the 700 MHz band".

**Data Availability Statement:** Not applicable.

**Conflicts of Interest:** The authors declare no conflict of interest.

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
