# Peer review of "Shielding Effectiveness of Unmanned Aerial Vehicle Electronics with Graphene-Based Absorber"

_electronics, doi:10.3390/electronics12183973_

Round 1

Reviewer 1 Report

In the present work, authors reported the shielding property of reduced graphene oxide (RGO) materials as an absorber in the frequency range from 100 MHz to 10 GHz. Results indicated that the absorber obtained the value of shielding effectiveness of 25 dB and 30 dB for 3 mm layer at 3.6 GHz and 5.8 GHz and of 29 dB and 39.5 dB for 5 mm layer at these frequencies respectively. Overall, this work has certain reference function. However, the synthetic process was not well provided, and some issues should be addressed.

1, The narrative in Abstract section was too tedious. The abstract should include the most important findings. Authors may focus on the what you have done and the related description should be refined to highlight your viewpoints. In addition, authors should figure out the significance and real-life application of the present paper.

2, The introduction writing part need to be improved. Also, the writing and presentation of the introduction lacks a bit in clarity. The paper requires some amount of rewriting to clarify all aspects of it, especially the novelty and new findings of this work that need to be clearly mentioned. Since numerous work regarding graphene and graphene oxide was reported, what is the difference of the present work? Authors should introduce the significance and originality.

3, How to obtain the GO materials? The synthesis process of GO was not introduced. Please, provide all product information including CAS Registry Number, density, size and others for all chemicals used in this study.

4, Please, provide all information of the methods of characterization such as XRD, SEM, FTIR, etc.

5, It was claimed that “ In fact, this diamagnetic phenomenon increases the absorption property of RGO, comparing to the simplified approach when permeability is taken as μ = 1. ” Please describe the statement.

6, Some key and important research results in graphene absorption field should be mentioned and cited so that we can provide a solid background and progress to the readers, such as Journal of Materials Chemistry C, 2016, 4, 9738; Advances in Colloid and Interface Science, 2020, 285, 102281; Scientific Reports, 2013, 3, 3421 .

7, It is unable to provide a satisfactory modelling of the absorption mechanism (s): To what is due microwave absorption (magnetism, polarization, relaxation, resonance ...)? This fundamental issue is not all answered.

8, Lots of articles have been already published on graphene absorbers. In such case, author should compare and discuss the previous results on graphene absorbers. Make a results Table of various parameters, preparation method, and peak absorption.

9, As we all know that the dispersion degree of graphene plays an important role in influencing performance of absorbers. Please discuss the influence of dispersion of graphene on SE.

Reviewer 2 Report

This article investigates the shielding performance of reduced graphene oxide (RGO) as an absorbent. By analyzing the shielding effect of RGO with different material layers in the typical frequency range, it can be proven that this material has low reflection coefficient and high absorption ability, which can protect drone electronic equipment and prevent system damage in the RF environment. It is a good candidate material for drone absorbers. In my opinion the work fits within the scope of Electronics and should be published after major revisions. In the following are several recommendations and clarifications that must be addressed before publication.

1.        The introduction lacks previous research on how to prevent harmful radiation and reduced graphene oxide (RGO). It is best to add some other studies and references.

2.        In order to observe the amplitude and phase of sample S parameters more intuitively, the S11 and S21 curves need to be provided.

3.        The authors state “After calibration the measurements were updated by the calibration coefficients.” The description of the calibration and measurement process of the sample is a bit vague.

4.        Legend and x-axis label are missing in figure 1.

5.        As the topic is about EM modulations, some recent works are suggested to be included in the introduction as following: [1]      L. Chen, Q. Ma, S. S. Luo, F. J. Ye, H. Y. Cui, and T. J. Cui, Small, p. e2203871, Sep 15 2022, doi: 10.1002/smll.202203871. [2] Q. Ma et al., eLight, vol. 2, no. 1, 2022, doi: 10.1186/s43593-022-00019-x.

Reviewer 3 Report

The article has a weak motivation, RGO can be used on any device for shielding. Using it on the UAV can be a good application but it is not enough for a publication. The rest of the article uses some conventional techniques which are not novel.  Moreover, the article is more focused on the material properties of the RGO, which of course can be found in the technical documents.  The article lacks any noval system model or any technique. There is no comparison with any benchmark methods. The article lacks any novelty in the methodology. 

The article needs some serious organization issues, and technical writing skills are weak.  The abstract is overlengthy with unnecessary explanations. 

Round 2

Reviewer 1 Report

.All issues were addressed, and this work can be accepted.

Author Response

Thank you for the positive opinion

Reviewer 3 Report

I have some of the concerns as follows

1. Practically can the authors highlight some cases of commercially available drones that are drowned due to base station (BS) RF radiations? why did you assume that the drone manufacturing industries have ignored such RF radiation threats (did they make the electronics of the devices with such flaws)?

2. How the authors can verify that the pure RGO method is the best as compared with other methods?  Did the authors provide any comparison graph for it? how can the authors verify that this is the best method as compared with the one already used in the manufacturing process in the industries?

3. If I assume the BS RF radiation is a threat to the electronics of the devices. Then there will be many devices with such threats. why UAVs are important among them? are UAVs manufactured carelessly with bad electronics?

4. How the industries will benefit from your findings? Such as DJI drones. did their drones drown near the BS?

English is fine.

Round 3

Reviewer 3 Report

Thank you for clarifying some of my concerns. I will suggest adding such discussion to the article as a motivation to help the readers understand why this study is essential.  I have no further comments.